# Investigation of Thermal Sensing in Fluoroindate Yb^3+^/Er^3+^ Co-Doped Optical Fiber

**DOI:** 10.3390/ma16062139

**Published:** 2023-03-07

**Authors:** Bartłomiej Starzyk, Gloria Lesly Jimenez, Marcin Kochanowicz, Marta Kuwik, Jacek Żmojda, Piotr Miluski, Agata Baranowska, Jan Dorosz, Wojciech Pisarski, Joanna Pisarska, Dominik Dorosz

**Affiliations:** 1Faculty of Materials Science and Ceramics, AGH University of Science and Technology, 30 Mickiewicza Av., 30-059 Krakow, Poland; 2Faculty of Electrical Engineering, Bialystok University of Technology, 45D Wiejska Street, 15-351 Bialystok, Poland; 3Institute of Chemistry, University of Silesia, 9 Szkolna Street, 40-007 Katowice, Poland; 4Faculty of Mechanical Engineering, Bialystok University of Technology, 45D Wiejska Street, 15-351 Bialystok, Poland

**Keywords:** fluoroindate glass, fluoroindate glass fiber, temperature sensing, InF_3_ glass, Er^3+^, Yb^3+^ ions

## Abstract

An investigation of fluoroindate glass and fiber co-doped with Yb^3+^/Er^3+^ ions as a potential temperature sensor was assessed using the fluorescence intensity ratio (FIR) technique. Analysis of thermally coupled levels (TCLs—^2^H_11/2_ and ^4^S_3/2_), non-thermally coupled levels (non-TCLs—^4^F_7/2_ and ^4^F_9/2_), and their combination were examined. Additionally, the luminescent stability of the samples under constant NIR excitation using different density power at three different temperatures was carried out. The obtained values of absolute sensitivity (0.003 K^−1^—glass, 0.0019 K^−1^—glass fiber ^2^H_11/2_ → ^4^S_3/2_ transition) and relative sensitivity (2.05% K^−1^—glass, 1.64% K^−1^—glass fiber ^4^F_7/2_ → ^4^F_9/2_ transition), as well as high repeatability of the signal, indicate that this material could be used in temperature sensing applications.

## 1. Introduction

Efficient temperature measurements are critical in the development of many fields including science, biomedicine, and industry. However, conventional contact thermometry has a slow response, high electromagnetic interference, low chemical resistance, among others [1,2]. For that reason, the interest in luminescence thermometry based on the fluorescence intensity ratio (FIR) technique has increased. This relies on the intensity ratio between two emission bands as a function of the temperature, which can be thermally coupled (200 cm^−1^ ≤ ΔE ≤ 2000 cm^−1^) or not. Compared to other systems, the FIR technique allows self-reference and avoids problems associated with measurement conditions (e.g., excitation intensity fluctuations, spectrum losses, etc.) [3,4], which increases the accuracy and sensitivity of the temperature measurement. This technique is typically used when trivalent rare earth (RE) ions are used as emission centers. Moreover, it is also possible to utilize others, for example, Ruby [5,6,7]. The advantage of RE ions is their unique optical and physicochemical properties, highlighting their multicolor emission, narrow emission/absorption bands, long decay times, and superior photostability. However, to achieve efficient emission intensity, a suitable matrix is essential. Among available matrices, the fluoride glass matrix, whose thermal stability is up to 583 K, excels for its lower phonon energy (~500 cm^−1^), superior mechanical properties, and high rare-earth ions solubility. Moreover, the low phonon energy of this matrix allows for achieving high emission intensity employing relatively low pumping power (<100 mW) or even using short optical fibers (1–10 cm), which could potentiate its use not only as a temperature sensor but also as a compact broadband source and lasers (here, the main interest is above 3 um) [8,9,10,11,12,13,14,15]. With such properties, the successful drawing of this glass could lead to an efficient and accurate temperature optical sensor. This is of particular interest because the fiber-optic sensors have higher sensitivity, superior resolution, flexible structure, compact size, and the ability to multiplex [16].

This work presents the up-conversion (UC) emission of a fluoroindate glass co-doped with Yb^3+^/Er^3+^ ions as a function of the temperature using the FIR technique of the thermally coupled levels (TCLs) and non-TCLs. As well as an initial fiber approach of this material as a potential element of a fluorescence fiber optic temperature sensor. The results showed that the ^4^F_7/2_→^4^I_15/2_ transition grows with the temperature, indicating an outstanding sensitivity to temperature changes which has not been observed before.

## 2. Materials and Methods

The preparation of both samples (glass and glass optical fiber) was performed in a glove box with a controlled atmosphere (O_2_, H_2_O < 0.1 ppm). Both were obtained using the following molar composition 35.8InF_3_-20ZnF_2_-20SrF_2_-16BaF_2_-4GaF_3_-2LaF_3_-0.8YbF_3_–1.4ErF_3_ (Yb^3+^: 1.73·10^20^ ions/cc and Er^3+^: 3.02·10^20^ ions/cc) employing just high purity reagents (99.99%). 

The glass was obtained employing the crucible melt-quenching technique. The mixed reactants were placed in a covered platinum crucible and fluorinated at 270 °C for 15 min. Then, the sample was melted at 900 °C for 5 min. After that, the liquid glass was poured out on a cold stainless steel plate, and the glass was relaxed at 290 °C for 2 h.

The glass fiber was fabricated using a modified crucible method. The melting glass ((µ = 10^9^ dPas)), obtained using the protocol described above, was extruded to obtain the core-rod preform. Then, the preform (µ = 10^4^ dPas) was drawn using the crucible nozzle, obtaining a fiber length of 20 mm.

The temperature-dependent luminescence measurements were performed on an electric heater at a 300–573 K temperature range. The material under assessment (glass or fiberglass), was placed over the stainless steel plate of the heater, then heated at X temperature (X = 300, 333, etc.), and remained there for 5 min to guarantee temperature stability. After that, the sample was irradiated at 980 nm (Pmax = 375 mW) with a CW fiber laser (Changchun New Industries Optoelectronics Tech. Co., Ltd., Changchun, China). The up-conversion measurements were performed using a 980 nm notch filter from Thorlabs with the number NF980-41 (Thorlabs Inc., Newton, NJ, USA). Their spectra were recorded employing different spectrometers for glass and fiber. A Zolix spectrometer (Zolix Instruments Co., Ltd., Beijing, China) equipped with an Omni-λ3007i (focal length—320 mm) monochromator and PMTH-S1 detector (185–900 nm) for the glass, and a Broadcom Qmini spectrometer (Broadcom Inc., San Jose, CA, USA) with a linear detection of the 2500 px CCD sensor for the fiberglass. JASCO V-670 UV-Vis NIR spectrophotometer was used to determine the spectrum of the absorption coefficient. The micrographs of the glass fiber were obtained using a Scanning Electron Microscope Phenom XL SEM with CeB6 source (ThermoFisher Scientific Inc., Waltham, MA, USA).

## 3. Results and Discussion

### 3.1. Fluoroindate Glass

The absorption spectrum of fluoroindate glass matrix co-doped with Yb^3+^/Er^3+^ exhibits the characteristic absorption bands from Yb^3+^ and Er^3+^ ions (Figure 1). It is evident that the absorption band centered at 976 nm, associated with ^2^F_5/2_(Yb^3+^) and ^4^I_11/2_(Er^3+^) transitions, fits well with 980 nm commercial lasers, indicating that the glass co-doped with Yb^3+^/Er^3+^ ions can be efficiently pumped with it.

To avoid the crystallization of the samples, the thermal studies were performed from 300 to 573 K, using steps of 30 K, since according to previous crystallographic studies, the crystallization starts above 583 K. Moreover, the glass-transition temperature (T_g_) is 580 K [13,14,15].

The up-conversion luminescence spectra of fluoroindate glass doped with Yb^3+^ and Er^3+^ ions at different temperatures are presented in Figure 2. These were split for a better resolution in the 475–505 nm range (Figure 2a). In Figure 2a a scale factor of 200 is used, compared to Figure 2b. The characteristic emission bands of Er^3+^ located at 492 nm (^4^F_7/2_ → ^4^I_15/2_), 523 nm (^2^H_11/2_ → ^4^I_15/2_), 545 nm (^4^S_3/2_ → ^4^I_15/2_), and 655 nm (^4^F_9/2_ → ^4^I_15/2_), are observed. However, the first one, typically negligible, was favored with the temperature indicating a high-temperature dependence. 

The up-conversion mechanism in the Yb^3+^/Er^3+^ system is based on an energy transfer process (inset in Figure 2b). Under the excitation of 980 nm, the sensitizer (Yb^3+^) absorbs the energy and the photons at the ^2^F_7/2_ (Yb^3+^) level are excited to ^2^F_5/2_ (Yb^3+^), then the energy is transferred to the activator (Er^3+^). Energy transfer (ETI) results in population of the ^4^I_11/2_ (Er^3+^) level (^2^F_5/2_(Yb^3+^) + ^4^I_15/2_(Er^3+^) → ^2^F_7/2_(Yb^3+^) + ^4^I_11/2_(Er^3+^)). After that, the Er^3+^ ions at the ^4^F_7/2_ level can be populated through two mechanisms. In the first one, Er^3+^ ions absorb energy by energy transfer (ETIII) from the ^4^I_11/2_ (Er^3+^) and ^2^F_5/2_ (Yb^3+^) levels (^2^F_5/2_ (Yb^3+^) + ^4^I_11/2_ (Er^3+^) → ^2^F_7/2_ (Yb^3+^) + ^4^F_7/2_ (Er^3+^)). The second mechanism is based on the interaction between two Er^3+^ ions, in which the populated ^4^I_11/2_ level of an Er^3+^ ion is excited to the ^4^F_7/2_ level by the energy transfer from the adjacent excited Er^3+^ ion (^4^I_11/2_(Er^3+^) + ^4^I_11/2_(Er^3+^) → ^4^F_7/2_(Er^3+^) + ^4^I_11/2_(Er^3+^)). As well as in ^4^F_7/2_, the population of the ^4^F_9/2_ level can be achieved through two mechanisms. The first one is based on the CR from ^2^H_11/2_ and ^4^S_3/2_. The second is energy transfer (ETII) from the levels ^4^I_13/2_ (Er^3+^) and ^2^F_5/2_ (Yb^3+^) (^2^F_5/2_ (Yb^3+^) + ^4^I_13/2_ (Er^3+^) → ^2^F_7/2_ (Yb^3+^) + ^4^F_9/2_ (Er^3+^)) [11,17,18,19,20].

This process is highly dependent on the photons’ behavior since an efficient upconverted emission mechanism relies on the ability to absorb and transfer them. However, this typical mechanism is influenced by the local temperature since a change in it could cause: (i) an increase in the non-radiative transitions or (ii) a population of higher energy levels through thermal excitation [21,22]. For that reason, the analysis of emission intensity as a function of the temperature has attracted growing interest. Among the techniques able to quantify such changes, the fluorescence intensity ratio (*FIR*) is maybe one of the most attractive since it relates the behavior of two emissions as a function of the temperature. This considers two energy levels that can be thermally coupled (TCLs) or not (non-TCLs). The energy levels are considered TCLs when the energy gap (Δ*E*) is between 200 and 2000 cm^−1^, which allows the population of the next higher energy level according to the Boltzmann distribution (Equation (1)) [23].
(1)$$FIR=\frac{I_{U}}{I_{L}}=Bexp\left(-\frac{\Delta E}{kT}\right)$$
where *I_U_* and *I_L_* refers to the integrated emission intensities of the energy transitions assessed (upper and lower), ΔE is the gap energy between them, *k* is the Boltzmann constant, T is the temperature, and B is an experimental constant. However, the Δ*E* restriction of the TCLs limits the sensitivity of the *FIR* technique. To overcome such limitation and considering that not only the emission intensity of the TCLs varies, Lu et al. proposed the assessment of other energetic levels known as non-TCLs which should be fitted following a polynomial equation (Equation (2)) [19]. In both cases, the absolute (*S_a_*) and relative (*S_r_*) sensitivity can be calculated using Equations (3) and (4). However, it is necessary to note that the main relaxation mechanism of non-TCLs is the multi-phonon relaxation and its analysis is complicated especially for the UC process [24].
(2)$${FIR}_{2}=a+bT+cT^{2}$$
(3)$$S_{a}=\frac{dFIR}{dT}=FIR\left(\frac{\Delta E}{{kT}^{2}}\right)$$
(4)$$S_{r}=\frac{1}{FIR}\frac{dFIR}{dT}\cdot 100\%=\frac{\Delta E}{{kT}^{2}}\cdot 100\%$$

Figure 3a presents the *FIR* between the TCLs (^2^H_11/2_ and ^4^S_3/2_), which fits well with linear regression; enabling to obtain the Δ*E* = 779.3 cm^−1^ through a simple linear regression, which slightly differs from the value obtained from the absorption spectra (Δ*E* = 744 cm^−1^), such a difference could be associated with the matrix employed or the accuracy of the measurements. While the *FIR* assessment between ^4^F_7/2_→^4^I_15/2_ transition (non-TCL), typically negligible, and (i) ^7^F_9/2_ (non-TCL), (ii) ^2^H_11/2_ (TCL), and (iii) ^4^S_3/2_ (TCL) are presented in Figure 3b–d, respectively. These show that the FIR increases with the temperature and fits well with the polynomial regression. The curve of *S_a_* and *S_r_* of each case is presented on the insets of Figure 3. All Sa increase with the temperature and reach their maximum at 573 K (Table 1). The highest one was 0.003 K^−1^, which corresponds to ^2^H_11/2—_^4^S_3/2_ transitions. While the *S_r_* achieves 1.04 K^−1^ at 300 K (^4^F_7/2—_^2^H_11/2_).

The obtained results indicate that the accurate selectivity of the matrix besides the ideal co-dopant concentration enables observing the high-temperature dependence of some negligible energy transitions, which can be considered in optical temperature sensing applications due to their outstanding sensitivity [19]. Compared to other promising results, the temperature dependence of ^4^F_7/2_→^4^I_15/2_ has not been reported before, even on the ZBLAN matrix under 377 nm excitation. These indicate that fluorinated glass is an ideal matrix for evaluating the temperature dependence of lanthanides, as its low phonon energy prevents non-radiative transitions and benefits typically insignificant transitions, making it an excellent candidate for temperature detection.

As repeatability is critical for sensing applications changes in the UC emission of ^4^S_3/2_ → ^4^I_15/2_ transition under constant NIR irradiation (980 nm), different pumping powers (200–375 mW), and different temperatures were investigated (Figure 4). The highest UC luminescence quenching degree (Δ*Rt*—Equation (5)) was 4% after constant irradiation for 300 s, with pumping power of 200 mW, and a temperature of 423 K. Moreover, the reversible luminescent switching capability of the sample under the same excitation but with a pumping power of 300 and 375 mW is illustrated in the inset on the right of Figure 4. Such results suggest that the material tested is attractive for sensor application.
(5)$${\Delta R}_{t}=\left(1-\frac{R_{0}-R_{t}}{R_{0}}\right)\cdot 100\%$$

### 3.2. Fluoroindate Optical Fiber

As the performance of fluoroindate glass co-doped with Yb^3+^ and Er^3+^ exhibited exceptional temperature sensing properties, an optical fiber was made with it. Then, its temperature-sensing properties were evaluated using the FIR technique employing the same transitions. Note that the surface area analyzed in each case is different since, according to scanning electron microscopy (SEM), the fiber has a diameter of ~100 µm (Figure 5c). Therefore, it is not surprising that, in general, the emission intensity of the fiber is lower (Figure 5). However, the temperature–emission dependence between ^4^F_7/2_→^4^I_15/2_ and ^4^F_9/2_→^4^I_15/2_ changes. In the first one, the influence of the temperature is not evident until the temperature exceeds 400K (Figure 5b), while in the second is not possible to observe a consistent temperature dependence since the intensity is down–up–down. 

The FIR between ^2^H_11/2_ and ^4^S_3/2_ at room temperature is ~0.15 in both samples (Figure 6a). However, the ratio between the glass and fiberglass changes with the temperature. On the fiber, the FIR at 573 K is ~26.2% lower than on the glass for ^2^H_11/2_—^4^S_3/2_, ^4^F_7/2_—^7^F_9/2_, and ^4^F_7/2_—^2^H_11/2_. The energy value between TCLs (^2^H_11/2_ and ^4^S_3/2_) is 620.2 cm^−1^ (Figure 6a).

The highest Sa of the TCLs (inset at Figure 6) was obtained at 428 K (0.0019 K^−1^), and the rest of the combination has a Sa around 0.00010 K^−1^. The maximum Sr of TCLs was achieved at 300 K (0.98% K^−1^) and, in the case of ^4^F_7/2_—^4^F_9/2_, ^4^F_7/2_—^2^H_11/2_, and ^4^F_7/2_—^4^S_3/2_, the Sr is 2.0, 0.8, and 2.0, respectively.

To assess the reliability of fiberglass, it was evaluated if FIR changes when the fiber is exposed to constant irradiation (Figure 7). Obtaining a maximum deviation of 8.3% when the sample was irradiated for 60 s with a pumping power of 300 mW at a temperature of 300 K. The reversible luminescent switching of the glass fiber confirms the viability of the material for temperature-sensing applications.

According to the sensitivity and consistency of the measurement of the glass and fiberglass, and taking into consideration the chemical stability of the matrix employed, it could be possible to use this material to fabricate an effective temperature sensor for use in immune locations and harsh environments [1,2,31].

## 4. Conclusions

Fluoroindate glass (wafer and fiber)co-doped with YbF_3_ and ErF_3_ has an outstanding temperature–emission dependence. According to the FIR technique, based on TCLs and non-TCLs, these materials have a high temperature–emission reliance, which was confirmed by their excellent sensitivity. The higher absolute sensitivity was 0.003 K^−1^ and 0.0019 K^−1^ for glass and glass fiber, obtained from the TCLs (^2^H_11/2_→^4^S_3/2_). While the superior relative sensitivity was obtained among ^4^F_7/2_—^4^F_9/2_ transitions (2.05% K^−1^ and 1.64% K^−1^ for glass and glass fiber). Taking into consideration the thermometric parameters and repeatability, this material could be used in designing a temperature sensor.

## Figures and Tables

**Figure 1 materials-16-02139-f001:**
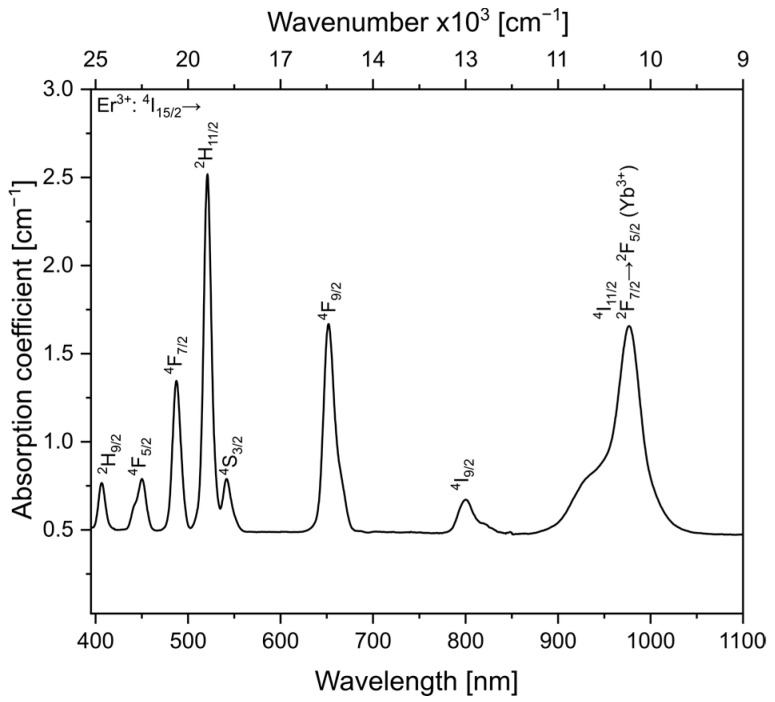
Absorption spectrum for the fluoroindate glass co-doped with Yb^3+^/Er^3+^ ions.

**Figure 2 materials-16-02139-f002:**
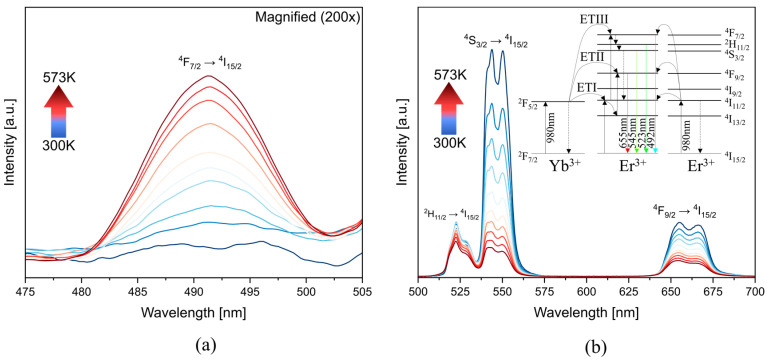
Emission spectra of fluoroindate glass co-doped with Yb^3+^/Er^3+^ under the 980 nm excitation at the (**a**) 475–505 nm range (**b**) 500–700 nm range (inset—Yb^3+^/Er^3+^ energy scheme).

**Figure 3 materials-16-02139-f003:**
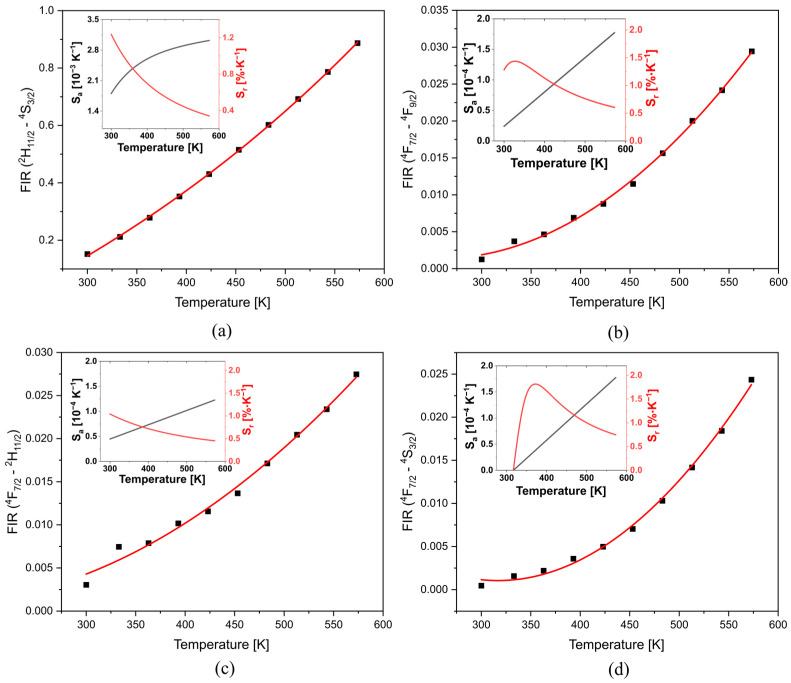
The FIR of (**a**) TCLs (^2^H_11/2—_^4^S_3/2_); (**b**) non-TCLs (^4^F_7/2—_^7^F_9/2_); (**c**) non-TCL—TCL (^4^F_7/2—_^2^H_11/2_); (**d**) non-TCL—TCL (^4^F_7/2—_^4^S_3/2_); Insets show the absolute and relative sensitivity.

**Figure 4 materials-16-02139-f004:**
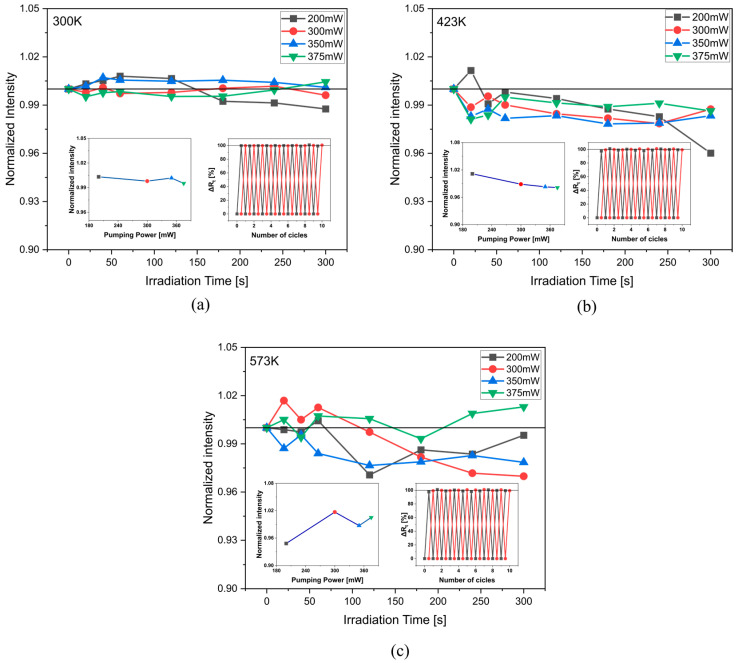
Normalized emission intensity at 545 nm versus irradiation time at (**a**) 300 K; (**b**) 423 K; (**c**) 573 K. The insets on the left show UC emission at 545 nm versus pumping power at 20 s irradiation time. The insets on the right show luminescent switching contrast of 980 nm excitation at 300 mW (black) and 375 mW (red).

**Figure 5 materials-16-02139-f005:**
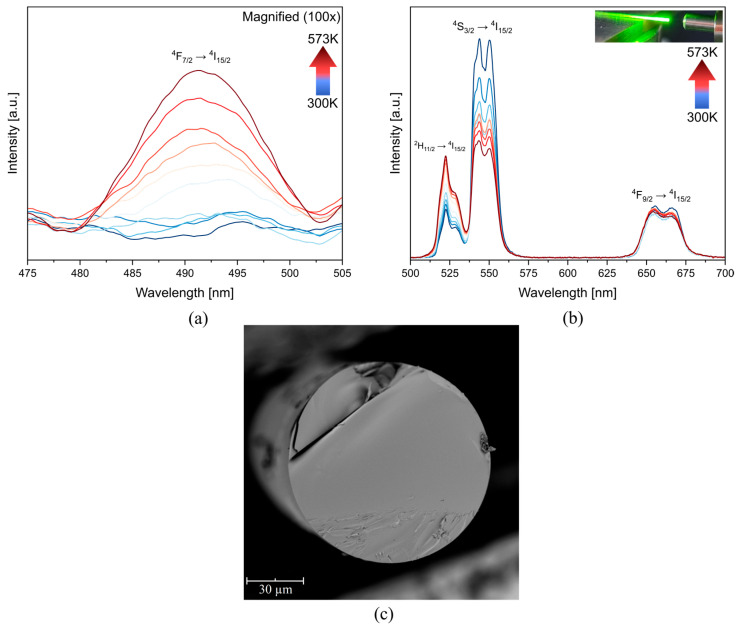
Up-conversion emission spectra of fluoroindate glass fiber co-doped with 0.8YbF_3_ and 1.4ErF_3_ under 980 nm excitation at the (**a**) 475–505 nm range (enlarge 100 times) (**b**) 500–700 nm range (inset—fiber under excitation) (**c**) SEM image of fiber cross-section.

**Figure 6 materials-16-02139-f006:**
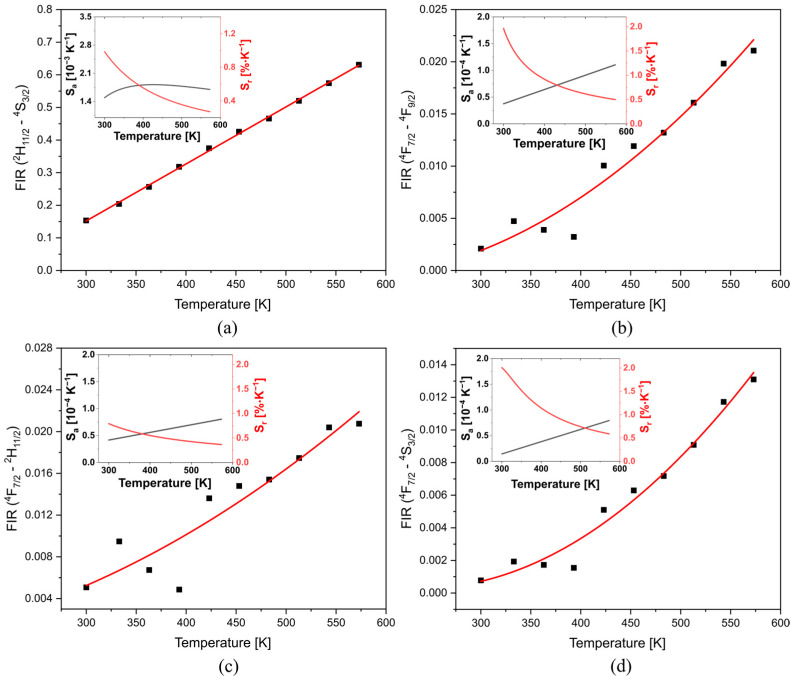
The glass fiber FIR of (**a**) TCLs (^2^H_11/2_—^4^S_3/2_); (**b**) non-TCLs (^4^F_7/2_—^7^F_9/2_); (**c**) non-TCL—TCL (^4^F_7/2—_^2^H_11/2_); (**d**) non-TCL—TCL (^4^F_7/2_—^4^S_3/2_). Insets show the absolute and relative sensitivity.

**Figure 7 materials-16-02139-f007:**
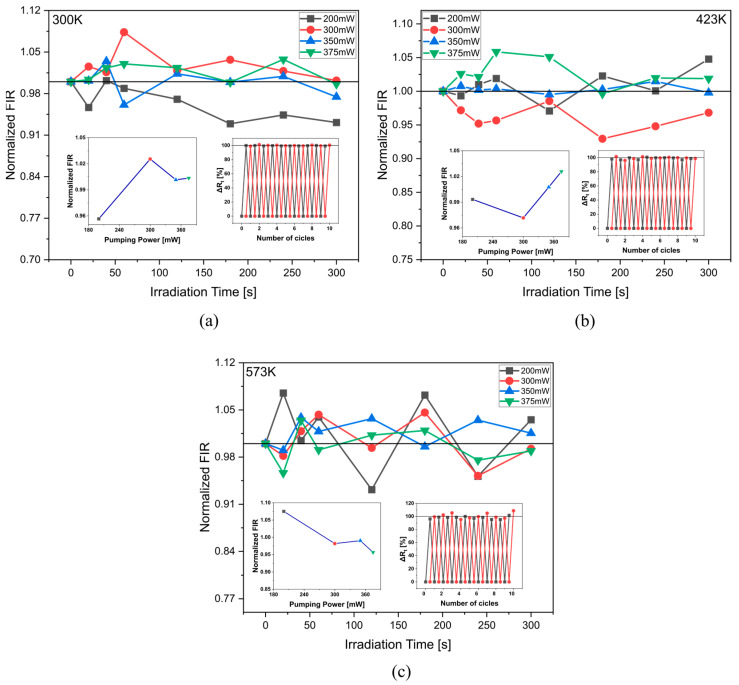
Normalized up-conversion FIR of glass fiber (TCL-TCL) versus irradiation time at (**a**) 300 K; (**b**) 423 K; (**c**) 573 K. The insets on the left show normalized UC FIR (TCL-TCL) versus pumping power at 20 s irradiation time. The insets on the right show luminescent switching contrast of 980 nm excitation at 300 mW (black) and 375 mW (red).

**Table 1 materials-16-02139-t001:** Optical thermometric parameters of fluoroindate glass and other glasses.

Glass	λ_exc_ [nm]	ΔE [cm^−1^]	Transition	S_a_ [10^−3^ K^−1^]	S_r_ [%K^−1^]	Ref.
InF_3_	980	780	^2^H_11/2—_^4^S_3/2_	3.00@573 K	1.24@300 K	This work
InF_3_	980	---	^4^F_7/2—_^7^F_9/2_	0.18@573 K	1.44@325 K	This work
InF_3_	980	---	^4^F_7/2—_^2^H_11/2_	0.12@573 K	1.04@300 K	This work
InF_3_	980	---	^4^F_7/2—_^4^S_3/2_	0.18@573 K	1.81@372 K	This work
TeO_2_-Al_2_O_3_-NaF-CaF_2_	980	767	^2^H_11/2—_^4^S_3/2_	7.45@548 K	1.25@298 K	[25]
Silicate glass	980	719	^2^H_11/2—_^4^S_3/2_	2.70@513 K	1.17@298 K	[26]
TeO_2_-BaF_2_-GdF_3_	980	771	^2^H_11/2—_^4^S_3/2_	6.84@548 K	1.25@298 K	[11]
TeO_2_-ZnO-ZnF_2_-La_2_O_3_	980	745	^2^H_11/2—_^4^S_3/2_	5.97@547 K	1.21@298 K	[27]
Fluoride glass	1480	769	^2^H_11/2—_^4^S_3/2_	4.00@548 K	1.25@298 K	[28]
Chalcogenide glass	1060	645	^2^H_11/2—_^4^S_3/2_	5.20@493 K	1.05@298 K	[29]
ZBLAN	337	732	^2^H_11/2—_^4^S_3/2_	4.60@527 K	1.19@298 K	[30]

## Data Availability

Not applicable.

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
