# Peer review of "Investigation of Thermal Sensing in Fluoroindate Yb3+/Er3+ Co-Doped Optical Fiber"

_materials, 2023, doi:10.3390/ma16062139_

Round 1

Reviewer 1 Report

In this paper, the author analyzed the characteristics of fluoroindate doped glass, and carried out temperature measurement experiments using the fluorescence intensity of the material, and obtained high temperature sensitivity. The following questions need to be further answered by the author.

1.     The author should introduce in more detail how to carry out the experiment through fluorescence effect. The experimental steps are lack of detailed description of the temperature measurement process.

2.     Compared with the traditional quartz fiber Bragg grating temperature measurement scheme, what are the advantages of this measurement scheme? How many times will the sensitivity be improved?

3.     Generally, the fluorescence measurement scheme is used for low-temperature measurement, and the advantages of this scheme should be explained in more detail.

4.     This fluorescence measurement scheme requires high stability of the light source. How to avoid the influence of light source fluctuation on the results?

5.     The author should explain in more detail the application value of this sensor and its outstanding advantages compared with similar sensors. Instead of simply listing the results, readers pay more attention to the significance of this work.

Author Response

Thank you for your comments and questions. Please see the attachment.

Reviewer 2 Report

1Page 2. Line 65 “…SAC-65 Fluo sample chamber with tunable laser excitation input at 980 nm…” What this phrase means?

2 Is the laser CW or pulsed?

3Page 2. Line 70 What is Tg? Please, define.

4Fig. 1. Energy level diagram is too small to read.

5Page 2. It is a mistake in notation 523 nm (2H15/2 4I15/2). It is 2H11/2.

6Page 3. Line 83-84. “To reach the higher level of 4F7/2 (Er3+), Er3+ ions absorb energy through energy transfer (ETII) from the 4I11/2 (Er3+) and 2F5/2 (Yb3+) levels.” Please, describe the process correctly.

7Page 3. Line 87. “The second is energy transfer (ETIII) from the levels 4I13/2 (Er3+) and 2F5/2 (Yb3+)”. Same as previous remark.

8Page 3, lines 96-99. ‘Transforming the Equation 1 to the form ln(FIR) depending on the reciprocal of temperature, made possible to determine the energy between: 2H11/2 and 4S3/2 (TCL - TCL) – 779.93±9.52 cm-1, 4F7/2 and 4F9/2 (non-TCL - non-TCL) –1293.15±55.66 cm-1 and 4F7/2 and 2H11/2 (non-TCL - TCL) – 859.63±58.65 cm-1.’ What are these energies mean? For example, the real energy gap between 4F7/2 and 4F9/2 level is ~ 5300 cm-1. Which energies you put into Eq. (2)? Please, explain in details what you are doing.

9What is a physical mechanism of changing the FIR for non-TCL levels of Er3+ with temperature?

1The dependence of the population of the Er3+ levels on pump power during up-conversion is nonlinear process and usually is very complicated. The rate equations must be written and solved to understand the change of population on pumping power accounting all the nonradiative and radiative rates of population relaxation for each level depending on temperature.

PPage 4 line 120. “This may be due to insufficient power to excite all available Er3+ ions.” This explanation is very strange and definitely incorrect.

1All the same comments for the fiberglass.

TThe paper in present form must be rejected.

Author Response

(The authors gave the same response as above.)

Reviewer 3 Report

The work is devoted to luminescence properties of fluoroindate glass and fiber, doped with Er3+ and Y3+ ions. The paper focuses on potential application of the material in temperature sensing applications. The paper is well-organized, the results obtained are well-formulated. However, there are several aspects that can be improved:

1. Choice of the material for the investigation is non-transparent. Introduction section does not have any information about luminescence properties of similar materials.
Was it the only material the authors had? Or was it chosen for some particular reason(s)?

2. It is unclear what advantages brings the use of the material (since there is no information on similar materials, see the previos paragraph)

3. Degree of doping by Er3+ and Y3+ ions is never given. It is possible for the readers to calculate it by their own from formula in line 56. But it would
be better to indicate RE ions concentration in ppm

4. Authors do not explain why fig.1 (also fig. 4) is divided into two parts. My suggestion is that they want to show the left part with higher resoution. Ok, then please specify it in the text.

Mistypes/mistakes to be fixed:

1. fundamental importance -> of fundamental importance (line 28)
2. an vital role -> a vital role (line 30)
3. despite limitations -> despite its limitations (line 31)
4. one of the best choice -> one of the best choices (line 34)
5. the obtained material was obtained -> the (desired) material was obtained (line 55)
6. melting process was happened -> melting was carried out (line 59)
7. according our previous results -> according to our previous results (line 69)
8. which it is known -> it is known (line 69)
9. range of to -> range from to (line 71)
10. is presented -> are presented (line 74)
11. energy transfer (et1) is transferred to populate -> energy transfer (et1) results in population  (line 82)
12. equation 1 were used -> equation 1 was used (line 90)
13. The non-TCLs levels were observed due to the increase accuracy of the analysis of the tested material
for sensor applications -> ????????????????????????????? (line 90 )
14. intensity -> intensities (line 94)
15. made possible -> made it possible (line 97)
16. (inset on right, Fig. 3) -> is illustrated by Fig.3, inset on right (line 123)
17. the obtained energy with values -> the energy values were obtained (line 145)
18. and was -> and amounted (line 151)
19. allow to state -> allows to state (line 172)
20. it possible -> it seems to be possible (line 176)
21. what is more -> moreover (line 186)

List of non-deciphered at first use terms and abbreviations

1. UC (line 47) up-conversion
2 T_g (line 70) glass-transition temperature

I think, after author's revision paper will be suitable for publication

Author Response

(The authors gave the same response as above.)

Reviewer 4 Report

In this paper, Yb3+ and Er3+ co-doped glass and fiber materials were prepared, and their temperature sensing properties were further studied. In addition, stability tests were carried out for different power, excitation and material irradiation time at three different temperatures. In this manuscript, some necessary characterization tests are still required for the prepared materials, and several confusing problems need to be explained. This article needs revisions before acceptance for publication. My detailed comments are as follows:

1. Is there any structural characterization to prove that the prepared material is glass?

2. The author claims that the fibers have been prepared, and the performance of glass and fibers is different due to the shapes and sizes. Therefore, the author needs to conduct SEM or other characterization and testing to explain the morphology, size and shape of the prepared fiber.

3. The excitation wavelength of the sample should be clearly indicated in the manuscript, and the excitation spectrum of the sample should be plotted.

4. This article explores three FIRs of this material, but by analogy, the combination of 4F7/2 and 4S3/2 (non-TCL-TCL) has not been explored. The author needs to supplement relevant data and give reasons why only three combinations have been explored in the article.

5. On the premise that data and analysis show that glass has better performance, why did the author choose to further prepare fiber? In the introduction, please write the reason for preparing fiber in this article, and describe the advantages of fiber materials in performance or application compared with other shapes and sizes.

6. The author needs to explain the deep reason why the performances of the prepared glass and fibers are different. How the shape and size of materials affect their temperature sensing performance in mechanism?

Author Response

(The authors gave the same response as above.)

Round 2

Reviewer 2 Report

1.      Page 1. Line 38. “This technique is typically employed when trivalent rare earth (RE) ions are used as emission center, nevertheless, it is possible to used others” English must be corrected.

2.      I do not understand the following. Please, give a figure for explanation.

“The other is the result of the cross-relaxation (CR) between the adjacent Er3+ ions in which one of them, located at 4I11/2 interacts with the other to gain energy allowing its movement to 4F7/2 (4I11/2(Er3+)+4I11/2(Er3+)4F7/2(Er3+)+4I11/2(Er3+)), also, with this mechanism (CR), the intense green emission of 2H11/2 and 4S3/2 levels is achieved.”

3.      The approach for analyzing the intensity of luminescence on temperature for non-TCLs is completely unclear. The main mechanism of relaxation of non-TCLs is the multiphonon relaxation. Its rate increases with the temperature. So, the intensity of fluorescence falls down with the temperature increase. You should analyze your data for non-TCLs according to the dependence of multiphonon relaxation rate on temperature. Please, see Optical Materials, 18, p. 355-365 (2002). An analysis of your data for non-TCLs in accordance with this paper will be correct physical approach.

Author Response

Thank you for your review. The answers are in the attachment.

Reviewer 3 Report

Now the paper looks fine.

Author Response

Thank you for your review.

Reviewer 4 Report

This paper can be accepted in current form.

Author Response

Thank you for your review.